# A regression-based method to estimate vessel mass for use in whale-ship strike risk models

**Alexandra Mayette** [1]*, **Sean W. Brillant** [1,2]

**1** Canadian Wildlife Federation, Halifax, Nova Scotia, Canada, **2** Department of Oceanography, Dalhousie University, Halifax, Nova Scotia, Canada

* alexandram@cwf-fcf.org

## Abstract

Vessel-whale collisions are one of the major threats that large whales face worldwide. The probability of lethal injury is an important factor to consider when conducting vessel strike risk assessment, which helps identify areas most at risk. To calculate the probability of lethality, these assessments have commonly used published models, which only account for the vessel speed. A biophysical model has been developed to simulate a vessel strike and estimate a probability of lethality, better accounting for the multiple parameters involved in the interaction, such as whale morphology, vessel dimensions, as well as speed. While the biophysical model is likely to provide more accurate estimates of lethality, some of the parameters required in the model input are not readily accessible, particularly a vessel's displacement (i.e., its total mass). To facilitate the use of the biophysical model for all users, our objective was to compute vessel-type-specific equations to convert a vessel's length overall to an estimated displacement. We collected the characteristics from n = 873 vessels, including vessel length overall, maximum deadweight tonnage, and total displacement when available. Vessel data was collected from Automatic Information System static data and other sources, and represented twelve vessel types. We computed a linear regression to test the correlation between the log of vessel length and the log of displacement, by vessel type. The model was significant (p < 0.001, $R^2 = 0.978$) and each vessel type was significantly different from at least one other vessel type. We demonstrated that these equations can then be used to compute the probability of lethality of a vessel strike with the biophysical model and obtain an estimate factored by vessel size. This can be an important tool for risk assessments and can better inform management and conservation measures for large whales.

## Introduction

Vessel strikes are a major conservation issue for marine mammals worldwide [1,2]. To better understand high-risk areas and inform management measures, modeling

---

**Data availability statement:** Data and R code are available in the following repository: Mayette, A. (2025). Data from "A regression-based method to estimate vessel mass for use in whale-ship strike risk models" [Data set]. Zenodo. https://doi.org/10.5281/zenodo.17782191.

**Funding:** This work was supported by Canada Nature Fund (2023-NF-MAR-016, granted to SB). The funders had no role in study design, data collection and analysis, decision to publish, or preparation of the manuscript.

**Competing interests:** The authors have declared that no competing interests exist.

methods often include the probability of a vessel strike to cause lethal injuries to a whale as part of the risk assessment [e.g., 3–8]. One of the determining factors of the probability of a lethal collision is vessel speed at the time of the strike [1]. A logistic regression describing the probability of lethality as a function of vessel speed was published by Vanderlaan and Tagart [9] and Conn and Silber [10], and these have been frequently used to calculate these probabilities. These models were computed from records of collisions with large whales around the world, with vessel speed and state of the injury known [9,10]. However, since then, new evidence has suggested that ship size could have an impact on lethality as well [11,12].

Kelley et al. [12] published a biophysical model simulating a perpendicular collision between a vessel and a whale, including parameters to account for vessel size: impact area ($m^2$), total mass (kg), as well as speed (m/s). The model is based on the physics of collision theory and calculates the stress from the deformation of the whale's four layers (skin, blubber, sublayer, and bone). The maximum stress measured during the simulation is then converted into a probability of lethality. The model can be configured for a specific whale species and vessel size by adjusting the biomechanical properties of the whale layers and the vessel dimensions. However, while the general parameters for the whale can be found in morphological studies and necropsy reports, the values for the dimensions of the ship are more difficult to find, particularly the total mass.

A recent study reviewed and updated the Conn and Silber [10] method with additional data from U.S. waters and tested the effect of other variables on the lethality of vessel strikes [11]. They concluded that speed, vessel size (four size categories based on vessel length), and whale taxon (humpback or not humpback) were significant variables to include in the calculation of lethality. The models of Kelley et al. [12] and Garrison et al. [11] each estimate the lethality of vessel strikes differently and have their own limitations. The biophysical model takes a mechanistic approach, using existing mathematical expressions to explain physical phenomena, but only estimates the lethality of a blunt force frontal collision and does not consider other types of injuries, like propeller laceration [12]. The logistic regression model of Garrison et al. [11] uses an empirical approach, statistically modeling observed data, but relies on available data and is currently constrained by external visible signs to classify the severity of injuries [11]. Garrison et al. [11] compared the results of both models and showed similar conclusions, highlighting that both models can be supplementary explanations of the impact of vessel strikes on whales.

These studies show that the lethality estimate of vessel strikes on large whales can be improved by accounting for the size of the vessels. Unfortunately, the necessary information is not always reported. Automatic Information System (AIS) is an international ship tracking system that, among other purposes, generates comprehensive data useful for vessel strike risk assessments. AIS comprises two types of information about the vessels: dynamic (position, speed, heading, etc.) and static (name, flag, vessel type, length overall, maximum deadweight tonnage, etc.). Of the three vessel parameters required for the biophysical model, speed is often readily available from the dynamic AIS data and the impact area can be generalized by

bow shape. However, the total mass of a vessel is not reported by AIS. Even the general specifications of a vessel rarely report the mass, with some exceptions for small vessels. In its current form, the biophysical model is not practical to use because of the disparity between the readily available data and the model input. The aim of this presented protocol is to facilitate the use of the Kelley et al. [12] biophysical model, as well as other current or future models that account for vessel mass, by developing vessel-type-specific equations to convert the length of a vessel to the mass of the vessel. Using available vessel specifications and naval architecture equations, we calculated linear regressions for twelve different types of vessels, describing the relationship between vessel length and mass.

## Materials and methods

### Ship characteristics terminology

We first define a few concepts concerning vessel characteristics and dimensions (Fig 1). Length overall (LOA) is the maximum length of a vessel's hull, parallel to the waterline. In the context of a vessel strike, the front of the hull, the bow, is generally the point of impact with a whale. Hulls and bows can have different shapes, depending on the vessel's use [13]. A common bow shape for small vessels is a flared bow consisting of two panels forming a vertical straight line, while larger commercial ships tend to have a bulbous bow, i.e., a bulb protruding at the bow to reduce wave resistance [13]. Displacement ($\Delta$) is the total mass of a vessel and is equal to the mass of water it displaces. Displacement is a formal characteristic of vessels given by the sum of the light-ship weight ($W_L$) (i.e., weight of the empty vessel including the weight of the structure, outfitting, and machinery) and the deadweight tonnage (DWT) (i.e., weight of the cargo, including the payload, fuel, crew, passengers and provisions, non-permanent ballast waters) [13]. The LOA and maximum DWT of a vessel are usually available in the static AIS data (in meters and tonnes, respectively), as they provide critical information for docking and guiding the maximum cargo allowed onboard. Another variable often reported as a characteristic of vessel size is the gross tonnage (GT). Although it could be confused with a unit of weight, gross tonnage is a measure of a vessel's volume capacity ($m^3$) [13] and therefore, is not suitable for estimating total mass.

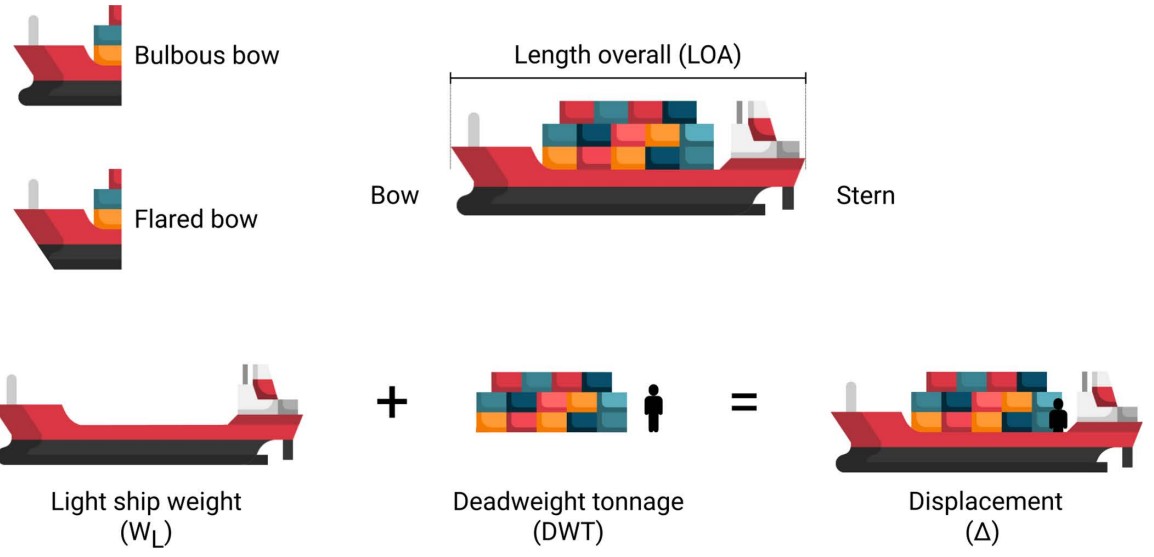

**Fig 1. Characteristics and dimensions of a vessel, used in this paper.**

## Calculating ship displacement

We used a subset of a static AIS dataset of vessels travelling in the Gulf of St. Lawrence, Canada (45–50.5° N; 58–68° W) between 2015 and 2022 (n = 724). Data was provided by the Marine Environmental Research Infrastructure for Data Integration and Application Network (MERIDIAN, Dalhousie University, Halifax, Canada). The data sample included a variety of vessel types, as the Gulf of St. Lawrence is a major waterway connecting the Atlantic Ocean to the Great Lakes, with many important ports along the way. In addition to commercial marine transportation, the area is used for fishing, aquaculture, oil and gas exploration, dredging, recreational boating, and tourism [14]. Vessels were categorized into twelve categories using general and detailed vessel type descriptions from the AIS: bulk carrier, container ship, cruise ship, ferry, fishing, government and research, passenger, pleasure craft, sailing, tanker, and tug (Table 1, Fig 2). Cruise ships and ferries were identified from the general category "Passenger" in the AIS data by further verifying the vessels' purposes and owners. The category "Other" included specialty vessels, crew boats, cable layers, drill shops, dredgers, etc. If two different vessels of the same type also had the same LOA and DWT (i.e., two vessels of the same model), only one replicate was kept.

As Canada only requires vessels of more than 20 m in length to use AIS (with some exceptions [15]), representation of small vessels (e.g., fishing, pleasure craft, sailing) was lacking in the AIS sample. To increase the sample size of smaller vessels in the data, we sourced additional vessel information online (n = 149, Table 1). Online sources included boat catalogues and ship brokers (e.g., itboat.com, www.nauticexpo.com, www.seaboats.net), motorized boat companies (e.g., Yamaha Motor Corporation, Lund Boat Company Inc), government websites, and boat databases (e.g., sailboatdata.com, Fishing Vessel Design Database [16]). These sources provided information on vessels from builders and manufacturers from across the world. Vessels were selected when information on LOA and displacement or net weight was available. When only the vessel's net weight was available, which would be the equivalent of the light-ship weight, we estimated the maximum displacement by adding the weight of an average person (75 kg) and one average luggage (20 kg) [13], multiplied by the maximum capacity of the vessel (i.e., maximum number of people onboard).

As mentioned above, static AIS data included LOA and maximum DWT for individual vessels. For vessels with known maximum DWT, we estimated displacement for each ship using the deadweight coefficient $C_D$, which is a ratio of DWT/Δ specific to vessel types [17], presented in Papanikolaou [13]. Relations between displacement and maximum DWT for additional vessel types were presented as linear or power regressions with associated equations in the appendices from

**Table 1. Summary of the sample of vessels used to measure the displacement, with the range of length overall (LOA), average LOA, average maximum deadweight tonnage (DWT) from the AIS data, and average displacement (kg) from data collected online. Associated standard deviation (SD) is presented.**

| Vessel type | Sample from AIS data | Sample from online sources | Sample size | Range LOA in m | Average LOA in m (SD) | Average maximum DWT (SD) | Average displacement in kg (SD) |
|---|---|---|---|---|---|---|---|
| Bulk Carrier | 100 | 0 | 100 | 85.9–300.0 | 221.0 (± 50.1) | 84 444 (± 61 877) | – |
| Container Ship | 100 | 0 | 100 | 84.0–336.6 | 240.7 (± 61.1) | 50 843 (± 29 726) | – |
| Cruise | 92 | 0 | 92 | 90.6–344.3 | 219.1 (± 69.7) | 6 059 (± 3 961) | – |
| Ferry | 47 | 0 | 47 | 25.5–203.3 | 100.1 (± 45.0) | 1 586 (± 2 022) | – |
| Fishing | 39 | 61 | 100 | 4.6–104.5 | 25.9 (± 22.2) | 794 (± 806) | 18 343 (± 41 154) |
| Government/ Research | 48 | 4 | 52 | 7.5–182.5 | 63.7 (± 31.1) | 1 617 (± 3 519) | 9 518 (± 6 559) |
| Other | 43 | 6 | 49 | 6.2–178.8 | 83.3 (± 45.3) | 5 284 (± 5 766) | 7 647 (± 5 287) |
| Passenger | 4 | 26 | 30 | 6.2–95.1 | 23.2 (± 20.9) | 609 (± 733) | 51 773 (± 106 519) |
| Pleasure Craft | 42 | 20 | 62 | 4.9–81.2 | 42.2 (± 23.0) | 286 (± 389) | 1 894 (± 1 105) |
| Sailing | 7 | 32 | 39 | 4.6–76.0 | 20.3 (± 18.2) | 512 (± 440) | 23 733 (± 45 241) |
| Tanker | 100 | 0 | 100 | 110.0–277.0 | 198.4 (± 44.2) | 63 904 (± 43 735) | – |
| Tug | 102 | 0 | 102 | 25.2–95.0 | 44.1 (± 18.6) | 952 (± 1 204) | – |

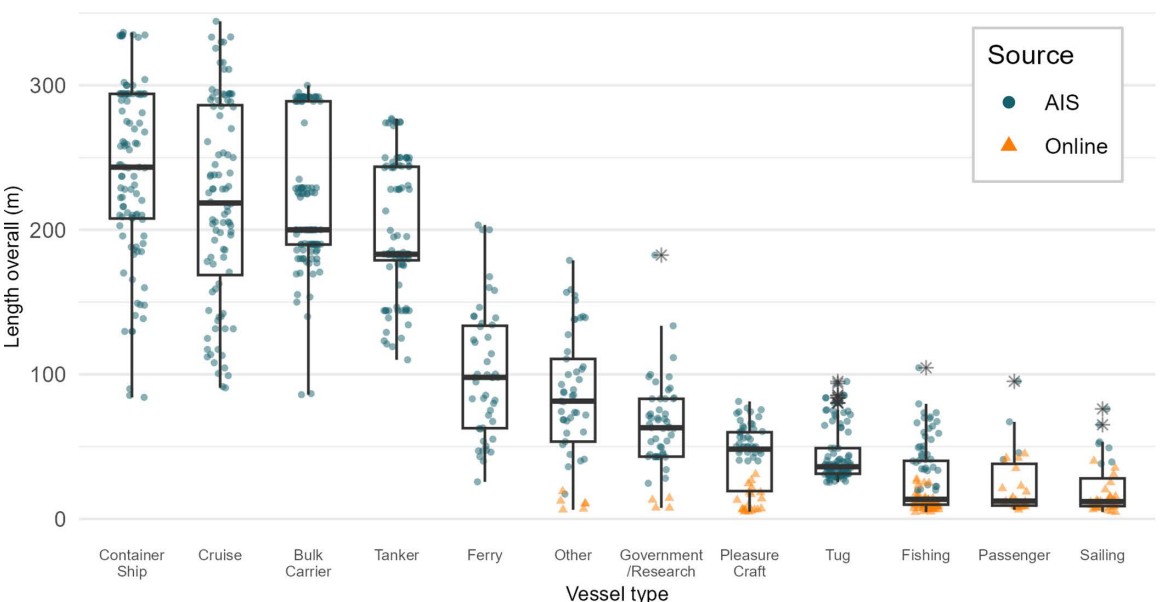

**Fig 2. Distribution of vessel length overall (m) by vessel type, ordered by the median, from the data set used in this study to find the correlation between length overall and displacement.** The dark circles represent samples from static AIS data and the orange triangles are samples from online sources.

studies by Kalokairinos et al. [18] and IHS Fairplay [19] (Table 2). The displacement calculation is described for each of the twelve vessel types in detail below.

**Bulk carrier.** Bulk carriers transport dry cargo loaded into the ship. They were identified from the cargo vessel general category, which was the most common vessel type in our AIS dataset. We randomly sampled 100 bulk carriers from our

**Table 2. Equations used to estimate vessel displacement (Δ) from deadweight tonnage (DWT) in tonnes (1000 kg), based on vessel type and average vessel length overall (LOA) in meters. Equations sourced from Papanikolaou [13] (Table 2.1 and appendices, referring to Kalokairinos et al. [18] and IHS Fairplay [19]) and expressed to isolate the displacement variable.**

| Vessel type | Equation | Reference |
|---|---|---|
| Bulk Carrier[a] | $\Delta = 2.21442 * DWT^{0.943855}$ | Kalokairinos et al. [18] |
| Container Ship[a] | $\Delta = 1.77955 * DWT^{0.975578}$ | Kalokairinos et al. [18] |
| Cruise (LOA ≈ 219 m) | $\Delta = 4.1152 * DWT$ | Papanikolaou [13] |
| Ferry (LOA ≈ 100 m) | $\Delta = 4.2918 * DWT$ | Papanikolaou [13] |
| Fishing (LOA ≈ 49 m) | $\Delta = 2.9940 * DWT$ | IHS Fairplay [19] |
| Passenger (LOA ≈ 62 m) | $\Delta = 5.988 * DWT$ | Papanikolaou [13] |
| Pleasure Craft (LOA ≈ 56 m) | $\Delta = 6.2893 * DWT$ | Papanikolaou [13] |
| Sailing (LOA ≈ 53 m) | $\Delta = 6.4516 * DWT$ | Papanikolaou [13] |
| Tanker[a] | $\Delta = 1.1213 * DWT + 4773.95$ | Kalokairinos et al. [18] |
| Tug | $\Delta = 5.2301 * DWT^{0.8752}$ | IHS Fairplay [19] |

[a]Some equations are in U.S. customary unit of tons (1016 kg).

dataset and information on the maximum DWT was available for all vessels. We calculated the displacement using the equation in Table 2. The displacement in U.S. tons was converted to kg.

**Container ship.** Container ships are another type of cargo vessel, transporting containers. We randomly sampled 100 vessels of this category from the AIS dataset. As DWT was available for all vessels, we used the equation in Table 2 to calculate the displacement and converted it from U.S. tons to kg.

**Cruise ship.** We had a total of n = 92 different cruise ships and all had maximum DWT information. The DWT/Δ ratio for large passenger vessels, such as cruise ships, is between 0.23 (LOA ≈ 200 m) and 0.34 (LOA ≈ 360 m) [13]. The average length of a cruise ship in our sample was 219.1 m, so we used the ratio 0.243 as the equivalent for an LOA ≈ 219 m (Table 2). The displacement in tonnes was converted to kg.

**Ferry.** A total of n = 47 unique ferries with a maximum DWT value were available in our data. The ratio DWT/Δ was between 0.16 (LOA ≈ 85 m) and 0.33 (LOA ≈ 120 m) for passenger vessels such as ferries, Ro-Ro (wheeled vehicle roll-on/roll-off), and Ro-Pax (roll-on/roll-off with passenger facilities) (Table 2). Since the average length of the ferry in our vessel sample was 100.1 m, we used the ratio 0.233 as the equivalent for an LOA ≈ 100 m. The displacement in tonnes was converted to kg.

**Fishing vessel.** In our data set, large fishing vessels (LOA > 20 m) often had maximum DWT available from the AIS, which resulted in n = 39 vessels. The DWT/Δ ratio for fishing vessels was given for a stern trawler and is between 0.30 (LOA ≈ 44 m) and 0.58 (LOA ≈ 82 m) (Table 2). Our average length for the fishing vessel type with DWT was 48.6 m, so we used a ratio of 0.334. The displacement in tonnes was converted to kg. To represent smaller fishing boats, we consulted the Fishing Vessel Design Database by the Food and Agriculture Organization of the United Nations [16]. This database provides fishing vessel designs from across the world, of boats built with various materials (e.g., wood, steel, glass-reinforced plastic) and ranging from 3.8 to 38 m in length. Of the 247 designs, 168 had the measure displacement available (in tonnes). We randomly subsampled 61 of those vessels to increase the representation of smaller vessels, for a total sample of n = 100, and the displacement value was simply converted to kg.

**Government/Research.** The category "Government/Research" included search and rescue vessels, survey and patrol vessels, icebreakers, and coast guard ships, and had a wide range of vessel lengths (average LOA = 63.7 m). We had n = 48 unique vessels of LOA > 20 m with available maximum DWT in our AIS data (average LOA = 68.1 m). We also found the displacement value of n = 4 other vessels of LOA < 20 m online, on national coast guard websites. To try capturing all types of vessels included in this category, we calculated the displacement values using the container ship equation, the fishing vessel (0.478 for LOA ≈ 68 m), and the small passenger vessel (0.176 for LOA ≈ 68 m) ratios and then averaged the value of the displacement. The displacement calculated with the container ship equation was converted from U.S. tons to kg, while the two other displacement values calculated were converted from tonnes to kg. The displacement value of the four vessels sourced online was directly available in kg.

**Other.** We had n = 43 vessels with maximum DWT from the AIS data and found n = 6 vessels of LOA < 20 m to better represent the smaller vessels. With an average length of 83.3 m, we used the same method as with the "Government/Research" vessel type. Considering the average LOA was 93.4 m for the vessels in the AIS sample, we calculated the displacement using the container ship equation, the fishing vessel (0.580 for LOA ≈ 93 m), and the small passenger vessel (0.212 for LOA ≈ 93 m). Displacement was converted from U.S. tons to kg for the container ship values and from tonnes to kg for the other two equations, before they were all averaged. For the vessel sourced online, some had the displacement value available directly in tonnes, which was converted to kg. For two vessels, the capacity was not available, so we only added the weight of one person and their luggage, as there would be at least one passenger onboard.

**Passenger.** The category "Passenger" had a small sample of vessels with available information on maximum DWT, as we removed the ferries and cruise ships (n = 4). The other passenger vessels included sightseeing tours, wildlife-watching tours, or vessels used as cargo and passenger vessels for remote communities. The average LOA of passenger vessels for the AIS sample was 62.2 m and 23.2 m for the whole sample. For the displacement calculation, we used a ratio of

0.167 for a length of 62.2 m, after the range of DWT/Δ ratio of 0.15 (LOA≈50) to 0.25 (LOA≈120 m) for small passenger ships (Table 2). The displacement in tonnes was converted to kg. For the vessels sourced online, displacement was already available or was calculated from the net weight corrected for maximum capacity.

**Pleasure craft.** The average vessel LOA for "Pleasure Craft" was 42.2 m. Vessels that had maximum DWT information available were usually large yachts (n=42), for an average LOA of 56.3 m and so we used the DWT/Δ ratio of 0.159 (LOA≈56 m) for small passenger ships (Table 2). We added n=20 smaller pleasure crafts (LOA<40 m) with the displacement or weight found online. The displacement in tonnes was converted to kg.

**Sailing vessel.** For sailing vessels, we took a similar approach to pleasure craft. For the vessel in the AIS data that had a maximum DWT value (n=7, average LOA=53.2), we used the ratio for small passenger ships (Table 2), 0.155 (LOA≈53 m). Since the sample size was quite small, we found n=32 other sailing boats with displacement available online. The displacement in tonnes was converted to kg.

**Tanker.** Tankers were frequent in the AIS data, and all had available maximum DWT. We randomly sampled n=100 tankers, giving an average LOA of 198.4 m. The equation in Table 2 was used to calculate the displacement, which was converted from U.S. tons to kg.

**Tug.** Tugs were also well represented in the data. We used all n=102 unique tug vessels with maximum DWT in the AIS data. We used the regression equation for offshore tug and supply vessels (Table 2). The displacement was converted from tonnes to kg.

After calculating a displacement value for each vessel and converting the displacement values to kg, we combined data from all vessel types. Based on the equations from Table 2 and after some data exploration, we suspected the relation between the predictive variable LOA and the response variable displacement would follow a power function. In addition, knowing ships are designed and built for specific purposes [17], we expected that different vessel types would have different length-mass relations and therefore, vessel type should be included as a predictive variable. We fit a linear regression on the log-transformed ship LOA (log(LOA)) with vessel type as an interaction term to predict the log-transformed displacement (log(Δ)). Model assumptions regarding outliers, normality, and homogeneity of residuals were verified following the Zuur and Ieno [20] guidelines and we used the significance level of $\alpha = 0.05$. To compare all vessel types to each other, we reran the model with each vessel type as the control group. To build more convenient equations describing the displacement from the LOA for each vessel type without the log transformation, we used the model coefficients of the linear function to convert to a power function, where the intercept, $b$, is transformed to the inverse log, $\exp(b)$, and the slope, $a$, is raised as the power of the predictive variable.

$$\log(\Delta) = a * \log(LOA) + b$$

$$\Delta = LOA^a * \exp(b)$$

## Application of the lethality model

To demonstrate the use of the regression-based method, we computed an example of converting the LOA of a vessel to its mass, to input in the biophysical model [12] and calculate the probability of lethality during a whale-ship strike. Using the "whalestrike" [21] R package version 0.5.1, we computed the lethality curve of six vessel types (container ship, cruise ship, ferry, pleasure craft, sailing, and tug), transiting at 10 and 20 kn (1 kn=1.852 km/h). We maintained the impact area variable constant, using the dimensions of a flared bow (Fig 1) (1.15 m x 1.15 m, [12]), for all six vessels. The lethality probability was computed for four different LOA (10, 20, 50, and 100 m). The default dimension of an average adult North Atlantic right whale (*Eubalaena glacialis*) was used as the whale model in the simulations.

All analyses were done in R version 4.5.1 (R Core Team, 2025), using the additional packages "tidyverse" [22] and "ggfortify" [23]. Data and R code are available in the supporting information.

## Results and discussion

The fitted linear regression showed a significant relation between the log(Δ) and the log(LOA), with vessel type as an interaction ($F_{23,849} = 1\,699$, $p < 0.0001$) (Fig 3). The adjusted $R^2 = 0.978$ demonstrated that the model explains most of the variability in the data. Model validation indicated no issues. Each vessel type was at least significantly different from one

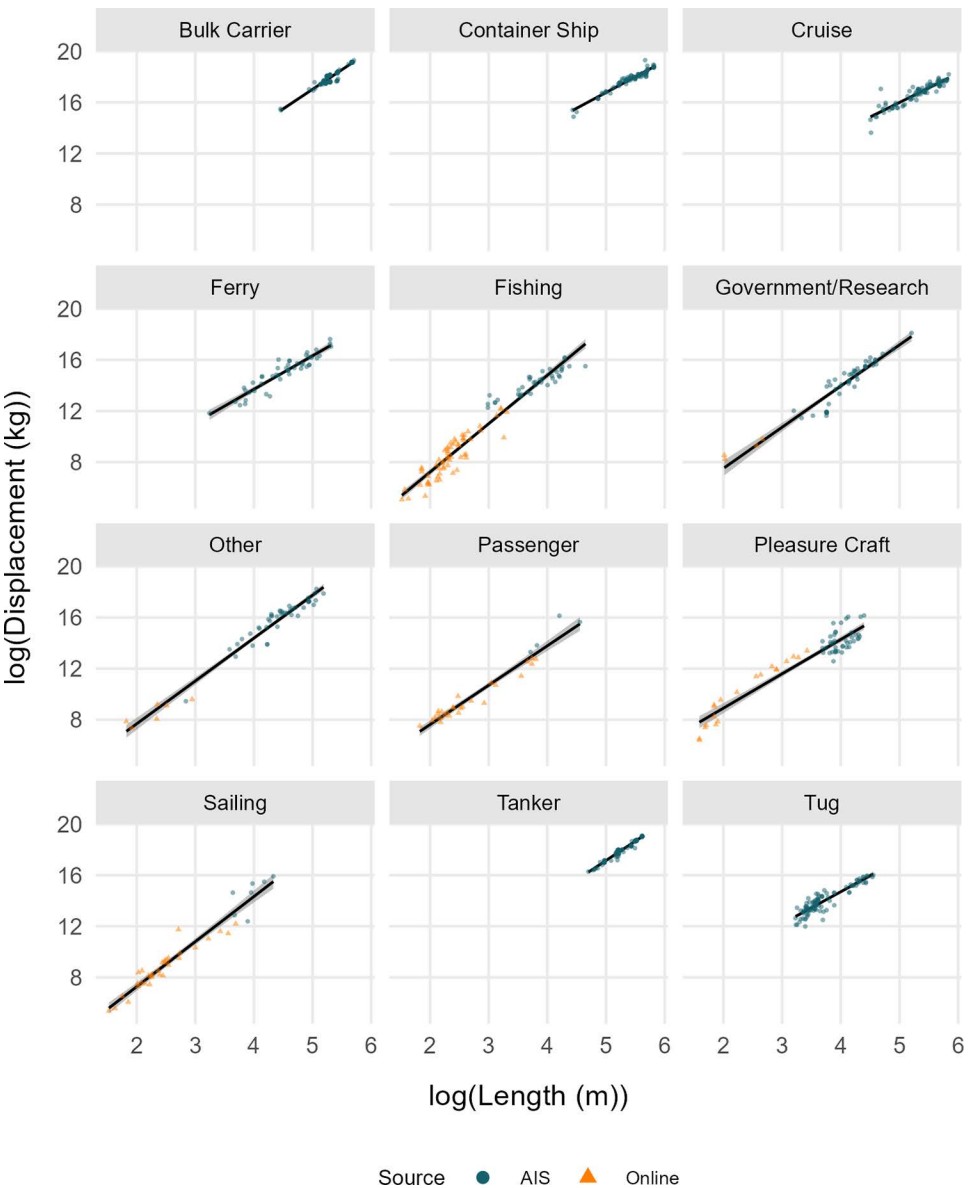

**Fig 3. Regressions of the log of vessel displacement (kg) by the log of vessel length overall (m), for each vessel type, with the 95% confidence interval shaded in grey.** The dark circles represent samples from static AIS data and the orange triangles are samples from online sources (when applicable).

other vessel type. Using the model coefficients, we described the equation to convert the LOA of a vessel to an estimate of total mass, for each vessel type in Table 3. Detailed results of the linear regression are available in the supporting information (S1 File), including the ANOVA table, the coefficients and their uncertainty, and a matrix comparing all vessel types.

The example of application of the method produced the probability of lethality for six vessel types, with four different LOA and going at two different speeds (Fig 4). Vessels of the same LOA and transiting speed showed a variation in the

**Table 3. Intercept and slope coefficients from the regression model, with associated standard error (SE). Final equations describing the ship displacement (Δ) (kg) with the length overall (LOA) (m), by vessel type as power functions.**

| Vessel type | Intercept (SE) | Slope (SE) | Equation |
|---|---|---|---|
| Bulk Carrier | 1.729 (1.117) | 3.058 (0.208) | $\Delta = 5.64 * LOA^{3.06}$ |
| Container Ship | 4.459 (0.917) | 2.461 (0.168) | $\Delta = 86.40 * LOA^{2.46}$ |
| Cruise | 4.580 (0.786) | 2.281 (0.147) | $\Delta = 97.51 * LOA^{2.28}$ |
| Ferry | 3.225 (0.691) | 2.619 (0.153) | $\Delta = 25.15 * LOA^{2.62}$ |
| Fishing | −0.344 (0.185) | 3.785 (0.061) | $\Delta = 0.71 * LOA^{3.79}$ |
| Government/Research | 1.083 (0.455) | 3.217 (0.112) | $\Delta = 2.95 * LOA^{3.22}$ |
| Other | 0.973 (0.364) | 3.354 (0.085) | $\Delta = 2.64 * LOA^{3.35}$ |
| Passenger | 1.463 (0.358) | 3.081 (0.122) | $\Delta = 4.32 * LOA^{3.08}$ |
| Pleasure Craft | 3.540 (0.261) | 2.685 (0.073) | $\Delta = 34.47 * LOA^{2.68}$ |
| Sailing | 0.2082 (0.298) | 3.529 (0.106) | $\Delta = 1.23 * LOA^{3.53}$ |
| Tanker | 1.981 (1.154) | 3.032 (0.219) | $\Delta = 7.25 * LOA^{3.03}$ |
| Tug | 4.649 (0.508) | 2.512 (0.136) | $\Delta = 104.48 * LOA^{2.51}$ |

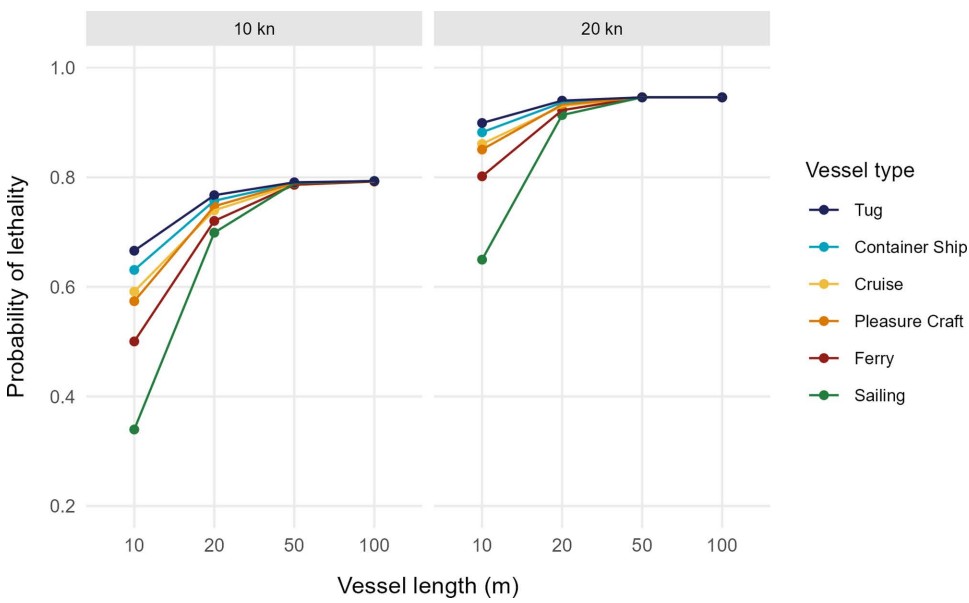

**Fig 4. The probability of lethality, calculated using the biophysical model, for six vessel types, with a flared bow, navigating at 10 and 20 kn, and with a length overall of 10, 20, 50, and 100 m.**

lethality due to the differences in displacement. These differences were more visible at lower LOA and converged at higher LOA. At a LOA = 10 m and a speed of 10 kn, the probability of lethality varied from 34% to 66% between the different types of vessels.

We proposed a substantiated method to estimate the total mass of a vessel from its length, and have provided unique equations for doing so for twelve different vessel types. It should be noted that because we used maximum DWT for these equations, the resulting calculations of displacement, therefore, correspond to the maximum vessel mass. In contrast, for vessels for which only the net weight was available, we estimated the maximum weight as best we could, based on the maximum passenger capacity, but there may be additional weight to the full displacement that was not accounted for.

Our classification of vessel types also generalized and aggregated the variety of vessel designs and purposes into a few categories. The use of a single equation (e.g., the stern trawler for all larger fishing vessels) may not accurately represent the relationship between LOA and displacement for all vessels in the category. Similarly, the deadweight coefficient was presented as a range for certain LOA values. We attempted to use a constant within that range that would best describe our sample based on the average length, but there could be some variation. It is also important to acknowledge that these are all estimates and have not been validated empirically. The method presented here was not intended to produce precise estimates of vessel displacement and should not be used in the context of ship design or naval architecture. Existing detailed marine engineering equations incorporate many other variables to estimate vessel mass that were not considered here, including beam, draught, and block coefficient of vessels.

Although vessels transiting the Gulf of St. Lawrence include both local and international vessels, our sample might not represent the full range and variety of ships designs. Specifically, with the vessels sourced online, data were limited to vessels for which the information necessary for this study was available and accessible in a language known to the authors. In terms of temporal differences, our AIS sample represented vessels built from 1953 to 2020, although the majority were built in the 2000s and 2010s. Vessels sourced online, particularly those listed on broker websites, varied in terms of construction year. However, there was likely a bias towards newer vessel models, especially for vessels sourced directly from merchant websites. Vessel designs have changed over time to be larger, more efficient, and adapted to new technology [24]. For this study, we did not have enough information to include the region of origin or year of construction for each vessel, and it is likely that other users of the biophysical model will not have this information either. Adding these variables could have led to more robust and complex equations to describe the relationship between LOA and displacement, accounting for some regional and temporal variation. However, the high coefficient of determination also indicates that the variables included in the model captured most of the variability in the data, and overall, these small variations might not have a significant impact on the final measure of the probability of lethality. In the interest of simplicity and given the difficulties involved in integrating this additional data, we believe that limiting the number of variables to a minimum, i.e., vessel type and length, makes the lethality model accessible to a greater number of users.

Calculating the probability of lethality is important for assessing the effect of vessel strikes on large whales, as well as the effectiveness of vessel management measures intended to reduce the lethality of strikes to whales. Using a model that estimates lethality more accurately will improve the assessment of risk and identification of high-risk zones. As shown by the example of lethality calculations, vessels of the same length do not necessarily present the same risk of lethality during a ship strike, due to functional and structural differences between ships that result in different masses. These differences in lethality are more evident for ships of smaller mass, as around 100 tonnes, the lethality increases only slightly with the increase of mass [12] (S1 File). This is explained by the mechanics of the biophysical model and the low variation in the effect of compression stress beyond this mass threshold [12]. However, it demonstrates the importance of accounting for these other variables and of the implications for management and protection measures for whales. For example, federal speed restrictions in the United States for the protection of the endangered North Atlantic right whale apply only to vessels 65 ft (19.8 m) and longer [25]. Fig 4 shows that

there is a large difference in the lethality of small vessels under 20 m and that vessels of this size, travelling at high speeds, still have a high probability of causing lethal injuries to an adult whale. Coastal areas where traffic is dominated by smaller vessels, whether commercial or recreational, could appear to pose a greater risk than previously estimated, given the mass and type of vessels now accounted for. Conservation measures should be adapted to the types of vessels, based on the risk they pose to their environment, in addition to size.

Our goal was to facilitate the use of a published model of vessel strike lethality on large whales [12] that accounts for the size of vessels as well as their speed. As the mass of vessels is not always reported, the utility of this model can be limited. We demonstrate a simple method to convert readily available data of a vessel (i.e., length) to a realistic estimate of its total mass that can be used in these more advanced models of lethality. If the vessel type categories presented in this study do not meet user needs for future vessel strike risk assessments, we suggest consulting the supplementary material (S1 File) to determine which vessel types are significantly different from each other and using only one regression equation in cases where they are not significantly different. Alternatively, if users cannot separate a more general vessel category into different categories (e.g., cargo ships into bulk carriers and container ships, or passenger ships into ferries and cruise ships), we suggest calculating the mass using each equation and averaging the results.

New and future vessel strike models that account for the size of vessels as well as their speeds [11,12] will allow researchers and managers to calculate more realistic estimates of the lethality of vessel strikes on whales. Accurate assessments of lethality are crucial for evaluating the effectiveness of vessel speed restrictions, which are increasingly used globally to attempt to reduce the harmful effects of vessel traffic on large whales.

## Supporting information

**S1 File. Results of the linear regression.**
(DOCX)

## Acknowledgments

We would like to thank and acknowledge Alexandra Cole and Shiva Jian-Javdan for their support within CWF's Marine Research team and MERIDIAN from Dalhousie University for helping us get access to AIS data. We would also like to thank Vince den Hertog from Robert Allan Ltd. Naval Architects & Marine Engineers for his feedback and expertise. Icons in Fig 1 were created by justicon and juicy_fish from Flaticon.com and modified by the authors.

## Author contributions

**Conceptualization:** Alexandra Mayette.

**Formal analysis:** Alexandra Mayette.

**Funding acquisition:** Sean W. Brillant.

**Investigation:** Alexandra Mayette.

**Methodology:** Alexandra Mayette.

**Project administration:** Sean W. Brillant.

**Supervision:** Sean W. Brillant.

**Visualization:** Alexandra Mayette.

**Writing – original draft:** Alexandra Mayette.

**Writing – review & editing:** Alexandra Mayette, Sean W. Brillant.

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
