## [Decision Letter · Decision Letter 0]

14 Nov 2025

Dear Dr. Mayette,

Thank you for submitting your manuscript to PLOS ONE. After careful consideration, we feel that it has merit but does not fully meet PLOS ONE’s publication criteria as it currently stands. Therefore, we invite you to submit a revised version of the manuscript that addresses the points raised during the review process.

We look forward to receiving your revised manuscript.

Kind regards,

Tien Anh Tran

Academic Editor

PLOS ONE

Journal Requirements:

Additional Editor Comments:

Thank you so much for your submission, After reviewing carefully your paper, there are some raised comments to improve the quality of your research. Kindly address all comments of reviewers then we are looking forward to receiving your revised paper.

Reviewers' comments:

Reviewer's Responses to Questions

**Comments to the Author**

1. Is the manuscript technically sound, and do the data support the conclusions?

Reviewer #1: Yes

Reviewer #2: Yes

2. Has the statistical analysis been performed appropriately and rigorously?

Reviewer #1: Yes

Reviewer #2: Yes

3. Have the authors made all data underlying the findings in their manuscript fully available?

Reviewer #1: No

Reviewer #2: No

4. Is the manuscript presented in an intelligible fashion and written in standard English?

Reviewer #1: Yes

Reviewer #2: Yes

Reviewer #1: Yes, the manuscript is technically sound, and the data support the main conclusion that vessel mass can be reliably estimated from LOA and vessel type using a regression-based approach. The statistical analysis is appropriately chosen for the research question. The authors apply a log-log linear regression with vessel type interactions to model the relationship between LOA and displacement, yielding a strong overall fit. However, the manuscript would benefit from greater transparency (e.g., reporting of coefficient uncertainty).

Please refer to attached word document for more in-depth suggestions and comments.

Reviewer #2: This study provides a fruitful refinement to the biophysical model elaborated by Kelley et al (2021) to estimate the lethality of a collision between a whale and a ship. Such biophysical approach is a potentially more precise way of assessing lethality of a strike than other approaches based on correlational approach as it permits better considering the characteristics of vessels (like size, draught) rather than simply the bessel category (cargo, tanker, sailing, etc). However, the Kelley model is not easy to implement due to the requirements of precise parameters to be known. The present work aimed at facilitating the access to one of these parameters, namely the vessel's displacement (mass). The authors collated the characteristics of a wide array of vessels from several sources and fitted linear régressions to provide user-ready equations converting the vessel length to vessel displacement. The paper is clearly written, the analyses are sound, the Supplementary Files are well constructed and insightful, and I only have a few minor comments to clarify some details in the text, that I detail below.

L141: a bracket is missing after the reference

L153: what is the rationale of only using a subset of vessels with LOA > 100 m? (I understand it is to equilibrate the sample size with vessels <100m but it may be good to explicitly state this)

L170: what are Ro-Ro and Ro-Pax?

L181: you use data from stern trawlers, yet fishing vessels are in essence quite variable in type, how this may affect the results ?

L252-261: what are the response vs predictive variables? what types of models were used, which packages and functions? More details are needed here about the modelling procedure.

L271-272: you should mention clearly in the text (probably at the end of the method) how were the equations in Table 3 built: it should be explained how coefficients were retrieved and converted back to natural scale - it would clarify that these equations are indeed user-ready and can be applied without any log-transformation.

**Do you want your identity to be public for this peer review?** For information about this choice, including consent withdrawal, please see our Privacy Policy

Reviewer #1: **Yes:** Raphael Mayaud

Reviewer #2: No

---

## [Author Response · Author response to Decision Letter 1]

3 Dec 2025

Reviewer 1 :

Thank you for taking the time to review our manuscript and provide constructive feedback. Following your recommendations, we have made some changes to the data used and provided more details on the methods and limitations to improve transparency. We have also included an example of the application of the biophysical model and added to the discussion the importance of considering vessel mass and type in ship strike risk assessment and management regulations.

First, based on your comments regarding potential regional bias in the vessel data, we revisited our selected data and changed some of it, particularly that from online sources. We paid closer attention to the origin of the vessels' builder or manufacturer. We tried to select a range of older and newer vessel models. In some cases, we were unable to find the original advertisement for a boat for sale (archived link), so we replaced it. We also found new sources of information, like the Fishing Vessel Design Database. These useful new resources allowed us to include vessels with a greater variety of designs, compared to primarily North American designs. For the AIS data, we didn’t make any major changes, but we realized that some vessels had replicates of the same model (e.g. a ferry or cruise ship with the exact same LOA and DWT but different ID), which we removed to have a greater variety of different ship models. We also decided not to set a length limit (i.e. including all vessels under a certain length, like we initially did with the cargos and tankers). All of this resulted in slight changes to the dataset, but overall, we hope that it has led to a more randomized and less biased dataset. The results of the regression model did not identify any problems with these changes, and we even improved the accuracy of the model.

Second, we added more detail on data selection, methodology and results. We also added a section in the discussion on limitations and potential variables not accounted for. We included the coefficients and their uncertainty for each equation.

Finally, following your recommendation, we added an example of the calculation of the probability of lethality. Using the equations to convert the LOA to total mass for six different vessel types, we showed how lethality can vary for vessels of the same length. We added some discussion on the implications of accounting for vessel mass and type in risk assessment and management. We referred to current management measures that could be improved by considering more vessel variables.

Overall, these changes did not alter the conclusions of this study, namely that these equations are necessary to use the biophysical model and measure the probability of lethality of ship strikes more accurately. The coefficients may have changed slightly, but with the statistical results, we are confident that the equations reasonably predict the relationship between length and mass.

Reviewer 2:

We thank you for taking the time to review our manuscript. We hope that the changes made help clarify some details in the methods and results.

See the Response to Reviewer file for more details on changes made to the manuscript.

---

## [Decision Letter · Decision Letter 1]

11 Dec 2025

A regression-based method to estimate vessel mass for use in whale-ship strike risk models

PONE-D-25-34309R1

Dear Dr. Mayette,

We’re pleased to inform you that your manuscript has been judged scientifically suitable for publication and will be formally accepted for publication once it meets all outstanding technical requirements.

Kind regards,

Tien Anh Tran

Academic Editor

PLOS One

Additional Editor Comments (optional):

Reviewers' comments:

Reviewer's Responses to Questions

**Comments to the Author**

Reviewer #1: All comments have been addressed

Reviewer #2: All comments have been addressed

2. Is the manuscript technically sound, and do the data support the conclusions?

Reviewer #1: Yes

Reviewer #2: Yes

3. Has the statistical analysis been performed appropriately and rigorously?

Reviewer #1: Yes

Reviewer #2: Yes

4. Have the authors made all data underlying the findings in their manuscript fully available?

Reviewer #1: Yes

Reviewer #2: Yes

5. Is the manuscript presented in an intelligible fashion and written in standard English?

Reviewer #1: Yes

Reviewer #2: Yes

Reviewer #1: I would first like to thank the authors for their response, having gone the extra mile by applying their concept as a worked example. The paper reads well and will be a good contribution to ship strike literature and risk assessments.

Reviewer #2: All comments have been addressed, and the updating of data has improved the analyses. The text is clear throughout, analyses sound, the paper is now ready for publication. It's a great work of high interest for anyone interested in mitigating collision risk in whales. Congrats!

**Do you want your identity to be public for this peer review?** For information about this choice, including consent withdrawal, please see our Privacy Policy

Reviewer #1: No

Reviewer #2: **Yes:** Charlotte Lambert

---

## [Editor Report · Acceptance letter]

PONE-D-25-34309R1

PLOS One

Dear Dr. Mayette,

I'm pleased to inform you that your manuscript has been deemed suitable for publication in PLOS One. Congratulations! Your manuscript is now being handed over to our production team.

Kind regards,

on behalf of

Professor Tien Anh Tran

Academic Editor

PLOS One